# Detecting Inference Attacks Involving Raw Sensor Data: A Case Study

**DOI:** 10.3390/s22218140

**Published:** 2022-10-24

**Authors:** Paul Lachat, Nadia Bennani, Veronika Rehn-Sonigo, Lionel Brunie, Harald Kosch

**Affiliations:** 1Univ Lyon, INSA Lyon, CNRS, LIRIS UMR 5205, F-69100 Lyon, France; 2Chair for Distributed Information Systems, Faculty of Computer Science and Mathematics, University of Passau, 94032 Passau, Germany; 3FEMTO-ST Institute/CNRS, Université Bourgogne Franche-Comté, F-25000 Besançon, France

**Keywords:** inference detection systems, sensor data, data privacy

## Abstract

With the advent of sensors, more and more services are developed in order to provide customers with insights about their health and their appliances’ energy consumption at home. To do so, these services use new mining algorithms that create new inference channels. However, the collected sensor data can be diverted to infer personal data that customers do not consent to share. This indirect access to data that are not collected corresponds to inference attacks involving raw sensor data (IASD). Towards these new kinds of attacks, existing inference detection systems do not suit the representation requirements of these inference channels and of user knowledge. In this paper, we propose RICE-M (Raw sensor data based Inference ChannEl Model) that meets these inference channel representations. Based on RICE-M, we proposed RICE-Sy an extensible system able to detect IASDs, and evaluated its performance taking as a case study the MHEALTH dataset. As expected, detecting IASD is proven to be quadratic due to huge sensor data managed and a quickly growing amount of user knowledge. To overcome this drawback, we propose first a set of conceptual optimizations that reduces the detection complexity. Although becoming linear, as online detection time remains greater than a fixed acceptable query response limit, we propose two approaches to estimate the potential of RICE-Sy. The first one is based on partitioning strategies which aim at partitioning the knowledge of users. We observe that by considering the quantity of knowledge gained by a user as a partitioning criterion, the median detection time of RICE-Sy is reduced by 63%. The second approach is H-RICE-SY, a hybrid detection architecture built on RICE-Sy which limits the detection at query-time to users that have a high probability to be malicious. We show the limits of processing all malicious users at query-time, without impacting the query answer time. We observe that for a ratio of 30% users considered as malicious, the median online detection time stays under the acceptable time of 80 ms, for up to a total volume of 1.2 million user knowledge entities. Based on the observed growth rates, we have estimated that for 5% of user knowledge issued by malicious users, a maximum volume of approximately 8.6 million user’s information can be processed online in an acceptable time.

## 1. Introduction

Nowadays, an increasing amount of individuals own wearable sensors (e.g., in smart phones or smart-watches) or ambient sensors that are deployed in their homes. The raising availability of raw sensor data motivates new services to offer insights about the health of their customers or about the energy consumption of appliances in their home. Sensor data collection is under the consent of the service customers and defined in a legal framework, such as the GDPR (http://data.europa.eu/eli/reg/2016/679/oj, accessed on 9 September 2022) (General Data Protection Regulation). In other words, the sensor data are collected for some given purposes.

Moreover, in recent years a diversity of data mining algorithms have been proposed in order to extract, from collected sensor data, information related to individuals. Although those algorithms can be used for genuine purposes, they can also be leveraged to gain insights breaching individuals’ privacy [1]. For instance, Banos et al. [2] and Chikhaoui et al. [3] demonstrated on the MHEALTH and the Opportunity datasets, respectively, that one can infer human activity from specific wearable sensor observations, if the observation interval exceeds a minimum temporal duration. Similarly, Cumin et al. [4] and Shahi et al. [5] demonstrated on the Orange4Home and the CASA-Aruba datasets, respectively, how daily human activities can be inferred from ambient sensors, thus considering spatial constraints. Aside from human activities, sensor data can be exploited to infer the behavior of individuals [6], the PIN (Personal Identification Number) code of a smart-phone [7], or to authenticate individuals based on keystrokes dynamics [8], etc. In any case, the human activities can be genuinely inferred for a purpose that the customers are consenting to (e.g., obtaining insights about the energy consumption of appliances, based on the human activities [9]), while being diverted to a non-consented use.

The diverted usage of mining algorithms to obtain more data on individuals is kindred to inference channels in the domain of databases, where it is defined as “the ability to determine sensitive data using non-sensitive data” [10], directly or indirectly as illustrated by the motivating example in [11]. Therefore, this usage can be considered as a kind of inference channel that we will call *inference attacks involving raw sensor data* (IASD) in the reminder of this paper. In the literature, the inference issue in relational databases is addressed via inference detection systems (InfDSs) [10,12,13,14,15,16]. To be able to detect inferences, the detection process needs to model the set of inference channels beforehand. Then, it gathers a knowledge about the data already queried by a user, and decides whether to answer the query or deny it in case of inference. In InfDS dedicated to detecting inference attacks in relational databases, inference channels can be universally represented by the same formalism (e.g., capturing functional dependencies (FDs) using Horn clauses [10]). However, inference channels involving raw sensor data have some additional specificities, such as temporal constraints which make existing modeling techniques not suitable.

The first requirement that an InfDS must meet for those kinds of inference channels is related to the representation of constraints. A mining algorithm is able to extract information from sensor data if constraints, with respect to those data, are satisfied. For instance, in order to mine human activities one must have access specifically to the acceleration and rotation of the wrist of an individual, on a time window of at least 2 s. The model of inference channels involving raw sensor data must enable the description of the considered sensor observation and its constraints (e.g., temporal for a time window, or spatial if the data must come from a specific room of a smart-home). Hence, the dynamic nature of sensor data queried by users must be captured as metadata related to the query, e.g., the quantity of data points queried for each observation, on which duration, data points that have been generated in which context, etc. Furthermore, a huge diversity of mining algorithms are proposed for different purposes. Each of them have distinct constraints. Consequently, the target model must be easily extensible in order to be able to capture this diversity of conditions related to the new discovered inference channels, as well as enabling an efficient reasoning on inference channels to detect when a user’s knowledge enable her to perform an inference attack. The InfDSs proposed in the literature model inference channels without considering constraints with respect to the data. Existing InfDS solutions are thus not suitable to model inference channels involving raw sensor data. Hence, the first contribution of this paper is the Raw sensor data-based Inference ChannEl Model (RICE-M), a new model which enables representing the conditions on sensor data, as well as the user knowledge gained by users querying sensor databases. Based on this model, we propose an InfDS, called RICE-Sy, which implements the detection of inference attacks at query-time (i.e., before the query is answered by the database [17]). We formally describe RICE-Sy’s principle and propose a proof of concept which demonstrates the feasibility of detecting inference channels involving raw sensor data. For this purpose, we use as a case study MHEALTH, the inference channel described by Banos et al. [2].

The second requirement is related to the capability of the InfDS to perform the detection at query-time while processing efficiently huge quantities of user knowledge. Sensors produce a continuous stream of data that are then stored in databases and performing the detection increases the total query answer time. Hence, the proposed InfDS should have an overhead as small as possible. The second contribution of this paper is a set of conceptual optimizations for RICE-Sy, which aim at reducing the quantity of user knowledge that must be considered when performing the detection on a query. Finally, we present two mechanisms to demonstrate the potential of RICE-Sy with respect to its detection time. We first describe strategies to partition knowledge gained by users which reduce the search space of the detection in RICE-Sy. Then, we introduce a new hybrid architecture based on RICE-Sy, called H-RICE-Sy, in which the detection at query-time is limited to users with a high probability to be malicious, in order to reduce the impact of the detection on the query answer time.

The remainder of the paper is organized as follows: Section 2 enumerates the limits of InfDSs proposed in the state of the art, with regard to our requirements. We introduce the inference channel that we use as a case study in this paper in Section 3. Section 4 provides the formal description of RICE-M. The generic workflow of RICE-Sy is provided in Section 5, followed by a set of conceptual optimizations that reduce the amount of data handled during the detection in Section 6. Section 7 presents the evaluation of the proposed conceptual optimization on the detection time of RICE-Sy. In Section 8 and Section 9, we present the partitioning strategies and the hybrid architecture H-RICE-Sy, respectively. We discuss the limitations of our solution in Section 10. Finally, Section 11 concludes the paper and presents future research directions.

## 2. Related Works

In the following, we first present the few related works associated with the detection of inference attacks stemming from sensor data, as well as highlighting the limits of the solutions, with respect to the requirements presented in Section 1. Noury et al. [18] propose an access and inference control model which enforces value constraints and temporal constraints on time series databases. The access control model they propose enables the definition of the rules which state the kind of permitted computation on time series and for which time interval. In their context, an inference attack occurs when at least two conflicting access rules lead to some temporal intersection which enable dishonest users to infer values that must be protected by the access control mechanism. Hence, Noury et al. do not propose a model to represent inference channels within the database. Furthermore, the proposed inference detection occurs at the design stage, i.e., before receiving queries on time series and it cannot operate at query time as required.

Due to the few works related to the detection of inference attacks involving sensor data, in the remainder of this section we present the InfDSs targeting personal databases protection only. Farkas et al. [17] show that two approaches are followed by InfDSs to prevent attacks. The first one is at design-time, where the schema of a relational database is modified to prevent the exploitation of inference channels. Turkanovic [19] shows that completely removing the inference channels is an NP-Complete problem. Furthermore, the schema of databases is not always available and even less modifiable. The second category of solutions acts at query-time. In this case, the InfDS detects when a user has queried enough knowledge to perform an inference attack. With respect to the requirements stated in Section 1, we will focus on InfDSs following this second approach.

Chen et al. [12] propose to tackle inference attacks by building first a Semantic Inference Model (SIM) representing the probabilistic dependencies between attributes. Then, a Semantic Instance Graph (SIG) is instantiated from the SIM to reflect its dependencies at instance level in the database and enrolled to detect the inference attacks. Their approach allows modeling of simple inference channels between attributes with some uncertainty limited to the conditional probabilities computed by the SIM. They use a common formalism, i.e., Bayesian Networks, to both model the inference channels and to keep track of the knowledge that users gain. The first limitation stem from the fact that the Bayesian Network is learned based on the data distribution of the database. Hence, this static approach implies that both the SIM and the SIG must be recomputed for each update (e.g., insertion of a new value) in the database, which is not practical in the context of a constant stream of generated sensor data. Moreover, random variables are not suitable to represent the metadata related to the issued queries since the probability distribution of those metadata can be computed only after all queries are issued. Hence, the SIM and SIG can not be leveraged to keep track of the knowledge of users and to reason on temporal or spatial constraints in the context of IASD.

Guarnieri et al. [13] propose a system where one module acts as a policy decision point whereas the other checks inference attempts. The latter relies on the security policy defining the inference threshold of sensitive information and the attacker model, which describes the user’s *a priori* knowledge (i.e., the inference channels). Then, based on the inference detection results, the first module decides whether to deliver the query answer or not. This approach allows modeling more general probabilistic inference channels than Chen et al. [12], since it is not only limited to dependencies between values of attributes, but can represent domain-specific relationships between the data items or the attacker’s prior knowledge. Similarly to Chen et al., Garnieri’s solution addresses inference detection with the strong assumption of static data in the databases which is not suitable for the continuously created stream of sensor data. Furthermore, Garnieri et al. focus on analyzing only closed queries issued to the database, whereas we assume that users issue queries to obtain sensor data points to then exploit data mining algorithms, thus metadata related to queries is not considered in this work.

El Mokhtari et al. [20] introduce an inference control mechanism which can be coupled with an access control mechanism. The authors model FDs as inference channels thanks to a matrix. In each row, the first column is a sensitive data that an owner wishes to keep private, and all the remaining columns describe the attributes that must be known in order to infer the sensitive attribute. El Mokhtari et al. propose to affect weights to the non-sensitive attributes to reflect their influence on the inference attack. The inference control mechanism keeps track of the data previously queried by each user to check if, combined with a new queried attribute, a user is able to infer a sensitive attribute. Hence, based on the weights of the accessed data, the proposed mechanism computes a percentage of completion towards an inference. When the completion reaches a given threshold, the query is rejected. Unlike Chen et al. [12] and Guarnieri et al. [13], the probabilistic representation of the inference channels relies on the weights affected to each attribute. It does not reflect the distribution of attribute values stored in the database that the mechanism protects. Moreover, the proposed model do not consider metadata related to the issued queries, since the exact queried attribute are modeled.

The system presented by Brodsky et al. [21] and extended by Toland et al. [10], models the FDs within a database, using Horn clauses, in order to compute the disclosed knowledge each time a query is issued. Then, based on the query log of the issuing user, the system either denies the answer or returns a fake one. To the best of our knowledge, the proposal of Toland et al. [10] is the only one that considers tuple updates (i.e., modification, deletion, or insertion) in the database by storing the most recent updates in the query history log. In their work, the FDs are limited to logical dependencies, thus probabilistic dependencies between data are not taken into account. They keep track and reason on the exact queried personal data, hence the modeled inference channels do not incorporate metadata related constraints. Moreover, since Toland et al. consider only personal databases, the dynamically of this solution has not been evaluated in the context of high frequency updates, such as the one produced by continuous insertion of sensor data.

Biskup et al. [14] propose a solution which protects logic-oriented information systems against inferences. The inference channels are here defined as propositional sentences in order to form an access policy. When a user queries information which appears within the policy, it is removed from the related sentences. Similarly to Garnieri et al.’s solution, Biskup et al. consider only closed queries in a logic-oriented information systems. When a sentence contains only a single remaining data, its access is restricted in order to prevent the inference attack. Hence, this solution does not keep track of the knowledge gained by users, but instead restrict the policy after each processed query. Therefore, neither the metadata of queries nor the temporal or spatial constraints can be modeled with this approach.

Staddon et al. [15] present a work based on cryptographic revocation techniques, where tokens are generated for objects (e.g., attributes or schema relations) that appear in an inference channel. All tokens of objects related to an inference channel are generated using the same single usage key. When a user queries an object, she needs to use the specific associated token which cannot be used afterwards. Thus, similarly to the solution of Biskup et al., the access to the object will be more and more restricted which prevents dishonest users to perform inference attacks since they cannot query all objects in an inference channel. This approach does not consider any constraint that the knowledge of users must satisfy to perform an attack. However, the most important limitation is that tokens can be generated at design-time since all the considered objects are known beforehand. In our context, an inference channel can be exploited via any sequence of queries while the resulting metadata satisfy the constraints. Hence, for a channel it is not possible to identify all the satisfying sequences of queries, thus tokens cannot be associated to metadata at design-time.

Qi et al. [22] aim at detecting and preventing inference attacks which occur when personal data are modeled as *resource description framework* (RDF) triplets. Their solution directly reasons on the set of triplets which are queried to check if a disclosure can occur. If so, the system can either deny to answer the query or delete triplets from the answer set in order to prevent the disclosure of sensitive data. This solution relies on domain specific ontology to perform the inference reasoning. To the best of our knowledge, ontologies do not exist to model and reason on temporal and spatial constraints related to metadata of queries on sensor databases. Furthermore, they consider the disclosure of sensitive triplets without keeping track of the knowledge that users gain by issuing queries. Hence, in addition to the lack of suitable ontologies, their solution is not suitable to model metadata about queries and to reason efficiently on knowledge gained via multiple issued queries.

In this article, we focus on protecting a single sensor database with an InfDS. Yet, a few InfDSs are proposed to detect inference attacks in a multi-database context, as described by Jebali et al. [23]. For the shakes of completeness we present, briefly, the limitations that also exist in those works with regard to our requirements. Both Chang et al. [16] and Lachat et al. [24] represent inference channels as probabilistic dependencies between attributes. They both have the same limitations as Chen et al.’s solution. Sellami et al. [25] focus on data integration systems where a unique view is provided to users by combining the results of a query issued to all databases. The authors consider simple FDs between attributes, their solution does not consider metadata related to the processed queries, nor temporal or spatial constraints that user’s knowledge must satisfy to exploit a FD. Jebali et al. [26] focus on data outsourcing and propose a solution which maintains access control policies of data owner while controlling inference leakage. To do so, they model the FDs graph corresponding to inference channels in the initial relational schema. When a user issues a query to the local database, the role-based access control mechanism checks if the user has the grant to access the queried data. If not the case, the history access control mechanism checks if the current query and the previous queries issued by the user can be leveraged to exploit a FD. If an attack is detected, the query is revoked.

Therefore, to the best of our knowledge, we observe that none of the existing InfDSs address the problem of detecting at query-time IASD. In the following sections, we first present the case study used to illustrate our contribution, we formalize our proposed model for inference channels based on raw sensor data, and we finally propose an InfDS based on this model.

## 3. Case Study: MHEALTH Dataset and Human Activity Inferences

The purpose of our case study is to model as an inference channel, an example of data mining algorithm applied to sensor data [2,3,4,5,6,7,8,9]. Hence, we need a detailed enough description of the mining process, in order to retrieve information related to the pre-processing of data. Those information represent the pre-requisite knowledge on sensor data that an attacker must have in order to infer then the mined personal data. Moreover, we aim to model a simple situation where the mined data are linked to a single individual.

In this paper, we consider as a case study MHEALTH, a public dataset published by Banos et al. [2] for recognition-based human activity. The authors describe the training process with good details, which enable us to extract the information required to model the inference channel. The dataset contains observations (such as acceleration, turn rate, etc.) from three wearable sensors placed on the chest, the right wrist, and the left ankle of a human being. Hence, the mined data are related to a single individual. Each sensor makes observations with a fixed frequency while a volunteer is doing each of 12 identified human activities (e.g., walking, cycling, etc.). The measures have been performed on 10 volunteers. The gathered sensor data are represented as a tuple of 23 attributes (e.g., x, y, and z axis for the acceleration) and the 24th attribute represents the label of the activity performed by the volunteer during the measurements. The authors aim to train a decision tree on this dataset in order to classify the measures into one of the activities. To do so, they consider 21 among the 23 attributes, since they judge that the two electrocardiogram measures from the chest sensor are not needed for a first evaluation. They use a non-overlapping sliding window with a duration of 2 s to pre-process the data. Thus, the constraint that must be satisfied to be able to exploit this inference channel is to know the values of 21 attributes for a common duration of at least 2 s. Then, one can infer which activity was performed by the volunteer who wore the sensors. The inferable knowledge is, here, the random variable activity, representing the probability distribution among the twelve identified actions as depicted by Figure 1.

## 4. RICE-M: The Raw Sensor Data-Based Inference ChannEls Model

We first summarize the assumptions related to the dishonest users, which are the potential attacker. Then, we describe and formalize the knowledge that the user obtains through queries and the inference channel description modeled by our proposed system. For readability purpose, the symbols presented in the formalization are listed and described in Table 1 and Table 2.

### 4.1. Assumptions Related to the Attacker Capabilities

In this section, we highlight the assumptions related to the dishonest users, and we point out the similarities and differences between the inference attacker model in the case of raw sensor data compared to the inference attacker when considering personal data. In both cases: (i) the attacker has agreed but controlled access to the data stored in the data sources he tries to query; (ii) to obtain the data he is interested in, the attacker queries the data he has access to. Doing so, he gathers knowledge that allows him to deduce non-accessed data; (iii) the attacker’s objective is to obtain user information of interest without querying it directly. This information could be sensitive data or non-sensitive data that allows him to obtain sensitive data; and (iv) the attacker owns external knowledge that guides him in asking the system for user information by means of queries.

The inference attack model in the case of sensor data differs from the attacker model on personal data by the following:Although the operation mode in the case of personal data remains the same: it is based on a knowledge of semantic links among data in a databases (e.g., the first and last name of an individual are both stored in two distinct databases) or it is based on the exploitation of the content of a database to deduce the value of an attribute (e.g., a functional dependency between the rank and the salary attributes can be exploited to infer the salary of a given individual), in the case of sensor data, the relationship between sensor data and personal data is a young and emergent research domain that uses machine learning techniques to be established [1], this external knowledge keeps increasing in diversity and volume.Due to the difference in data nature, in the case of sensor data, the attacker will try to obtain a certain amount of data of the same nature (e.g., value of some attribute during a period of time) whereas in the case of personal data, he will simply ask for the value of an attribute.

### 4.2. Modeling User Knowledge

Independently from data nature, the inference detection system stores user knowledge reflecting the information that has been already queried to be able to detect inference attacks. However, to take into account sensor data specificities, i.e., the temporal and spatial constraints, as well as the continuous nature of the generated sensor data, we have proposed an extension of user knowledge modeling explained in the next section.

To exploit an inference channel involving raw sensor data, an attacker queries the targeted data from a stream. For the sake of simplicity, we assume that queries follow the SQL-inspired theoretical grammar defined in Figure 2. It is currently focused on the temporal constraint and it can be extended to more selection conditions when generalizing the model. Data streams registered in the protected data stream management system (DSMS) are denoted by the set *S*. Each stream s∈S is composed of a set of attributes denoted by $A_{s}$ corresponding to sensor attributes. Thus, the set $A={⋃}_{s\in S}A_{s}$ of all attributes measured by sensors describes all the queryable attributes in the sensor database. Finally, accessing an attribute can be restricted to temporal intervals $(t_{b}$, $t_{e})\in T,\;t_{b}\leq t_{e}$ (i.e., the beginning and ending timestamps). Detecting inferences at run-time implies that an InfDS reasons on the accessed data of each user u∈U having issued at least one query on the database. Let MK represent the set of all the metadata describing data extracted by the users via queries. The smallest metadata extracted is called a Metadata Knowledge Unit (MKU) and defined as a tuple mku=〈a,c〉∈MK where a∈A and c∈T is a selection condition, and whose meaning is that the user has asked for the attribute *a* where the condition *c* is true, e.g., $c=(t_{b}=1,t_{e}=4)$. Each query issued by a user *u* will be represented in the user knowledge by $Q^{u}=\left\{mku_{1}^{Q^{u}},\ldots ,mku_{n}^{Q^{u}}\right\}\subseteq MK$. For instance, the query SELECT $a_{1},a_{2}$ FROM *s* WHERE INTERVAL $\left(t_{b},t_{e}\right)$ issued by user *u* leads to the extraction of the metadata set $Q^{u}=\left\{\langle a_{1},\left(t_{b},t_{e}\right)\rangle ,\langle a_{2},\left(t_{b},t_{e}\right)\rangle \right\}$. To keep track of the MKUs of each user, the system relies on a Query History Log (QHL). Formally, the log is a function QHL:U↦MK mapping a user *u* to their respective set of MKUs extracted after issuing a set of queries at an instant t. For instance, after issuing 5 queries, the log of a user *u* is $QHL\left(u\right)={⋃}_{i=1}^{5}Q_{i}^{u}$.

Focusing on the inference channel of our case study MHEALTH which is based on a temporal duration, an attacker will aim to query data points related to the 21 required sensor data for a common temporal interval of at least 2 s. For example, the user queries SELECT $a_{1},\ldots ,a_{21}$FROM mhealth_stream WHERE INTERVAL (1,4) issued by a user *u* results in $Q^{u}=\left\{\;\langle a_{k},\left(1,4\right)\rangle |1\leq k\leq 21\;\right\}$.

### 4.3. Modeling Inference Channels

Each new discovered channel has its specificities in terms of implied attributes and conditions under which these attributes could be exploited. For example, for the MHEALTH inference channel, the targeted attributes are $a_{1}$ to $a_{21}$ and the channel could be exploited if the user detains the values of all of them during 2 s.

In the inference channel modeling, each channel has a unique identifier i∈I and is described by a tuple 〈P,F,X〉∈ICD, where ICD represents all channels known by RICE-Sy. The set P is composed of a set of patterns $\left\{p_{1},\ldots ,p_{h}\right\}$. A pattern corresponds to an expected shape of MKU, for instance a MKU referencing a specific attribute for a specific type of condition. Patterns are functions, such that $\forall p_{i}\in P,\;p_{i}:MK\mapsto MK_{p_{i}},\;1\leq i\leq h$, and $MK_{p}={⋃}_{p\in P}p\left(MK\right)$. The set P⊆P,|P|>0 describes the patterns of MKUs which must be known by users to be able to satisfy the channel constraints. These constraints encode the logical goal which must be satisfied by the MKUs of a user (e.g., from the QHL(u) and a new query $Q^{u}$) to reach an inference. Thus, the set $F=\left\{f_{1},\ldots ,f_{l}\right\}\subseteq F,\;l\geq 1,\;\forall f\in F,\;f:MK_{P}\mapsto \left\{⊤,⊥\right\}$ represents the constraints of a specific channel, while the set F represents all the constraints which can be defined for any known inference channel. Finally, the set $A_{inf}$ denotes the inferable attributes of each channel that are not queryable from the data stream ($A_{inf}\cap A=\emptyset$), i.e., the attributes known by a user when performing the described inference. Any attribute $a_{inf}\in A_{inf}$ has a given domain $dom(a_{inf})$ which represents values associated to this attribute. Each inferable attribute is modeled by a random variable *X*, such that $\forall a_{inf}\in A_{inf}:X:dom\left(a_{inf}\right)\mapsto \left[0,1\right],\;X\in X$. The set X contains the inferable attribute knowledge of each inference channel. X∈X represents the percentage of confidence, as a probability distribution, of each value a user *u* obtains when exploiting an inference channel with their QHL. Finally, the description of each inference channel is stored within the Inference Channel Repository (ICR), formally defined as ICR:I↦ICD. The description of an inference channel *i* is retrieved via $ICR\left(i\right)=\langle P_{i},F_{i},X_{i}\rangle$. We assume that the modelled inference channels are provided to the ICR module by the administrator of the system. Hence, for a channel i∈I and a user u∈U, the relation between $P_{i}$, $F_{i}$, and $X_{i}$ is that if $\exists MK_{P_{i}}^{u}={⋃}_{p\in P_{i}}p\left(QHL\left(u\right)\right):MK_{P}^{u}\neq \emptyset \wedge {⋀}_{f\in F_{i}}f\left(MK_{P}^{u}\right)$ then *u* knows $X_{i}$, i.e., which means that user *u* succeeds to infer $X_{i}$.

Focusing on our case study, we denote the related inference channel by i=1. This channel is related to the 21 attributes values known for a temporal interval, which lead to the following patterns $P_{1}=\left\{\;p_{k}|p_{k}\left(MK\right)=\left\{\;mku|\forall mku=\langle a^{\prime },c^{\prime }\rangle \in MK:a^{\prime }=a_{k}\wedge c^{\prime }\in T\;\right\},\;1\leq k\leq 21\;\right\}$. The single constraint $F_{1}=\left\{f\right\}$ means that these attributes must all be known for a common interval of at least 2 s. Thus, it is defined by $f\left(MK_{P}\right)=\exists \left(t_{b_{ref}},t_{e_{ref}}\right)\in T:\left[t_{e_{ref}}-t_{b_{ref}}\geq 2s\right]\wedge \forall \langle a,\left(t_{b},t_{e}\right)\rangle \in MK_{P}:\left[t_{b}\leq t_{b_{ref}}\leq t_{e_{ref}}\leq t_{e}\right]$. Finally, when a subset of a user’s MKUs matches the patterns and satisfies the constraint, they obtain the inferable knowledge $X_{1}$, i.e., the random variable modeling the t$activity\in A_{inf}$ where dom(activity)= {Walking, …, Cycling}, as depicted by Figure 1.

## 5. Generic Workflow of RICE-Sy

The architecture of our RICE-Sy, illustrated in Figure 3, is composed of two main parts. The *knowledge base* maintains the QHL of each user and the ICR contains known inference channels descriptions. We assume those descriptions to be provided by a third-party entity. For example, the administrator of a RICE-Sy instance can review the scientific literature describing mining process on data from the same domain as the data protected by the instance. Each time a new inference channel is identified, the administrator models it and adds it to the ICR. The *reasoner* detects inference occurrences based on the content of the knowledge base. The QHL size grows every time a query is processed whereas the ICR size increases only when new channels are modeled in the system. The system maintains the following invariant status: the QHL of any user does not satisfy the conditions of any inference channel stored in ICR. This can be formally expressed as $\forall u\in U,\;\forall i\in I,\;MK_{P}^{u}={⋃}_{p\in P_{i}}p\left(QHL\left(u\right)\right):MK_{P}^{u}=\emptyset \vee \exists f\in F_{i}:\neg f\left(MK_{P}^{u}\right)$. Thus, considering the QHL of a user at any time, the only case where the user can reach the inference channel conditions is by receiving a query $Q^{u}$, such as $Q^{u}$ content, added to the user’s QHL allows them to reach the inference channel conditions. The query processing by RICE-Sy is the following (cf. Figure 4): the MKUs contained in $Q^{u}$ are first extracted and sent to the system ❶. Then, the *pre-processing* module retrieves all the MKUs from the QHL(u) ❷ and produces the set $MK_{Q\cup QHL}^{u}=Q^{u}\cup QHL\left(u\right)$ ❸. The *detection* module, second module of the reasoner, checks if a subset of MKUs in $MK_{Q\cup QHL}^{u}$ allows the user *u* to exploit one of the modeled inference channels ❹, in which case the detection system notifies the database system sending a binary value ❺, otherwise, the user receives the data they asked for and their QHL is updated ❻ by inserting the MKUs of $Q^{u}$ into QHL(u).

### 5.1. Detection Step

The detection module is formalized as a function detection:MK↦{⊤,⊥} which searches within the set $MK_{Q\cup QHL}^{u}$ the MKUs satisfying each constraint $f\in F_{i}$, for each inference channel i∈I. This detection step is described in Algorithm 1 assuming that there is no collusion between users.

**Algorithm 1**: Detection step of our RICE-Sy.

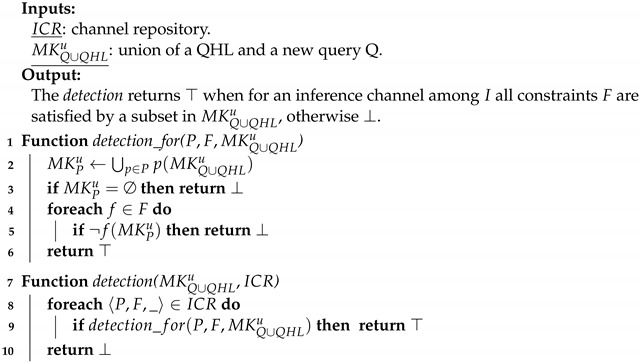



For example, focusing on the inference channel of our case study, a first query issued by the user u∈U: $Q_{1}^{u}=\left\{\;\langle a_{i},\left(1,4\right)\rangle |1\leq i\leq 11\;\right\}$ targets the first 11 observations required to exploit the channel 1. Since it is the first query from a user *u*, QHL(u) is empty, so $MK_{Q\cup QHL}^{u}=\left\{Q_{1}^{u}\right\}$. For this first query, no pattern in $P_{1}$ could be satisfied as attributes values for $a_{12}$ to $a_{21}$ are missing in $MK_{Q\cup QHL}^{u}$. Consequently, this first query does not lead to an inference. When no inference is detected, the QHL is updated with the new MKUs from the query. In case *u* issues a second query $Q_{2}^{u}=\left\{\;\langle a_{i},\left(1,4\right)\rangle |12\leq i\leq 21\;\right\}$ targeting the remaining observations, since $QHL\left(u\right)=\left\{Q_{1}^{u}\right\}$ then $MK_{Q\cup QHL}^{u}=Q_{1}^{u}\cup Q_{2}^{u}$. This subset is then identified as matching the patterns of the inference channel and satisfies its unique constraint, thus leading to an inference.

In terms of complexity, the worst case occurs when all the subsets of MKUs within $P(MK_{Q\cup QHL}^{u})$, i.e., the set of all possible subsets of $MK_{Q\cup QHL}^{u}$, are valid subsets for each modeled channel. In this case, the constraints of each inference channel must be applied to the subsets, leading to a general complexity of $O(|P\left(MK_{Q\cup QHL}^{u}\right)|\times \sum_{i\in I}\sum_{f\in F_{i}}O\left(f\right))$. Moreover, in terms of storage, after issuing a few queries, the size of each user’s QHL grows quickly. This implies that in the detection algorithm $|MK_{Q\cup QHL}^{u}\left|\approx |QHL\left(u\right)\right|$ in the long term. The time computation will grow quadratically based on the size of the QHL, which will thus quickly harm the data availability. To reduce detection time, a first algorithmic optimization is presented in the next section and aims at reducing detection time.

## 6. Steps of the Conceptual Optimizations

According to [27], selection queries on time series databases are executed in 80 ms on average. Performing the detection of inference at query time implies increasing the query answer time. In order to let RICE-Sy action time acceptable, we assume that the average detection time must take at most as long as the query execution itself. To reach this objective, we present in this section a set of conceptual optimizations.

As described in Section 5, the role of the pre-processing module is to compute the set $Q^{u}\cup QHL\left(u\right)$ that serves as input to the detection module, each time a query is issued. We recall the invariant status of the QHL, when a new query is processed the detection module needs only a subset of QHL(u) to check if an inference is occurring or not. Hence, limiting the size of the detection input drastically reduces the time needed for detection. The main idea of the proposed optimization steps is to retrieve the minimum relevant subset of MKUs in the QHL. This is obtained thanks to the three steps, depicted in the *pre-processing* module in Figure 4, and detailed below. Note that the proposed optimizations target only MKUs with temporal constraints.

### 6.1. Optimized QHL

As a first conceptual optimization, we propose to reduce the number of MKUs stored in the QHL. In order to detect inference attempts, the detection module consolidates the input MKUs. This process is defined by the function consolidate:MK↦CMK⊆MK which takes as input a set of MKUs and searches sequences $\left\{\langle a_{i},c_{1}\rangle ,\ldots ,\langle a_{i},c_{m}\rangle \right\}$ where $a_{i}\in A$ and the *m* conditions can be merged into a unique condition $c_{1m}$ equivalent to $\left\{c_{1},\ldots ,c_{m}\rangle \right\}$. Furthermore, the system consolidates QHL(u) with $Q^{u}$ for MKUs related to the same attributes, assuming that the DSMS pre-processes each query so that MKUs within $Q^{u}$ are already consolidated. The QHL is now defined by the function OpQHL:U↦CMK and referred to as the Optimized Query History Log (OpQHL). To compute OpQHL(u), the consolidation relies on the rules illustrated in Figure 5. Algorithm 2 formally describes the consolidation module. The worst case occurs when the set $A^{Q^{u}}$ of attributes referenced by MKUs in $Q^{u}$ is equal to the set of attributes referenced by MKUs in OpQHL(u). The consolidation step processes every MKU in $Q^{u}\cup OpQHL\left(u\right)$, resulting in a time complexity of $O(|A^{Q^{u}}\left|\times (\right|Q^{u}\left|+|OpQHL\left(u\right)|)\right)$.

**Algorithm 2**: Consolidation module for selection conditions.

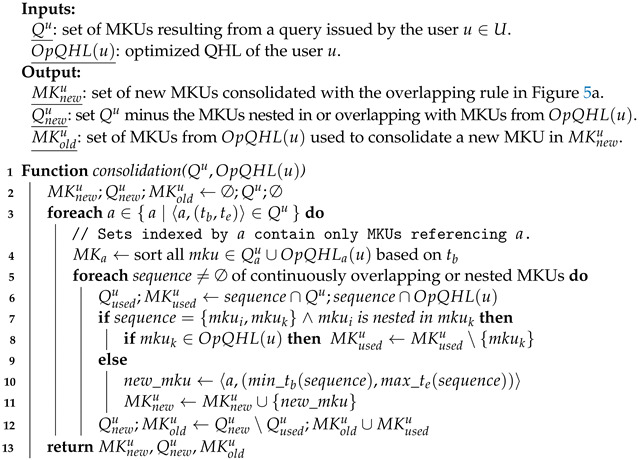



### 6.2. Query Based Filtering

Only the MKUs from a query and from a QHL which are referencing the same attribute and are nested or overlapping, as defined in Figure 5, are needed for consolidation. When consolidating QHL(u) with a query $Q^{u}$, it is, thus, not necessary to take the whole set OpQHL(u) as an input. For instance, when receiving $Q_{2}^{u}$ of the example, cf. Section 5.1, the query-based filtering step will retrieve from the QHL only MKUs referencing $a_{12}$ to $a_{21}$ which are overlapping or nested in the temporal interval (1,4), if any. Thus, filtering only this subset from OpQHL(u) enables the consolidation module to reason on the relevant MKUs, and thus reduces the detection time. This filtering is formally defined as a function $query\_based\_filtering:MK_{Q\cup OpQHL}\mapsto MK_{cons}$ in Algorithm 3. In the worst case $MK_{cons}^{u}=OpQHL\left(u\right)$ where each MKU in the OpQHL is related to a MKU in the query, in which case the time complexity of this filtering is $O(|Q^{u}|\times |OpQHL\left(u\right)|)$.

**Algorithm 3**: Filtering modules for the consolidation and the detection.

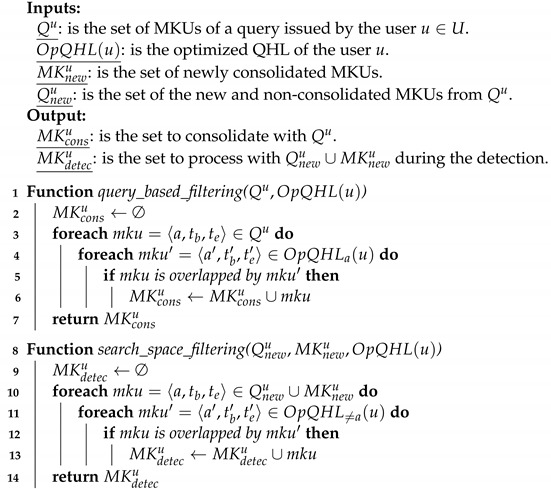



### 6.3. Search Space Filtering

Since the system invariant states that an inference can occur only when incorporating the MKUs of a query $Q^{u}$ with the MKUs of the OpQHL(u), the detection can limit its search space to the subset of MKUs from the OpQHL(u) overlapping with the new MKUs in $Q_{new}^{u}\cup MK_{new}^{u}$ which are related to a different attribute. Taking again the query $Q_{2}^{u}$ given in Section 5.1, the search space filtering step will retrieve from the QHL only MKUs referencing $a_{1}$ to $a_{11}$ which are overlapping or nested in the temporal interval (1,4), if any. This filtering is formally defined as a function $search\_space\_filtering:MK_{Q_{new}\cup MK_{new}\cup OpQHL}\mapsto MK_{detec}^{u}\subseteq OpQHL\left(u\right)$ in Algorithm 3. The worst case occurs when all the attributes referenced by MKUs in $Q^{u}$ are not referenced in OpQHL(u) and all MKUs stored in OpQHL(u) are overlapped by MKUs in $Q^{u}$. Then, $MK_{detec}^{u}=OpQHL\left(u\right)$ leading to a time complexity of $O(|Q_{new}^{u}\cup MK_{new}^{u}|\times |OpQHL\left(u\right)|)$.

### 6.4. Optimized RICE-Sy Workflow

To summarize, cf. Figure 4, once receiving a new query $Q^{u}$ issued by a user *u* ❶ the *query based filtering* extracts from OpQHL(u) ❷ the subset of MKUs $MK_{cons}^{u}$ ❸ which can be consolidated with $Q^{u}$. The *consolidation* will compute ❹ the set $MK_{new}^{u}$ of the new consolidated MKUs, the set $Q_{new}^{u}$ of MKUs from $Q^{u}$ which did not lead to a new consolidated MKU or are not nested in a MKU from $MK_{cons}^{u}$, and, finally, the set $MK_{old}^{u}$ containing the MKU from $MK_{cons}^{u}$ which lead to a new consolidated MKU. The *search space filtering* extracts from OpQHL(u) ❺ the subset of MKUs $MK_{detec}^{u}$ which must be processed with $Q_{new}^{u}\cup MK_{new}^{u}$ ❻ by the *detection* module in order to check if a user *u* attempts to exploit one of the inference channels described in the ICR ❼. In case no inference is detected, OpQHL is updated ❽ by first removing all the MKUs which have been used to consolidate new MKUs (i.e., the set $MK_{old}^{u}$) to finally insert MKUs obtained thanks to the consolidation (i.e., $MK_{new}^{u}$) and via the query (i.e., $Q_{new}^{u}$), for MKUs which have not been consolidated. By storing the consolidated MKUs, the system is able to reduce the size of the OpQHL in the long term and maintains the invariant status stated before.

## 7. Evaluation of the Conceptual Optimizations

To demonstrate the feasibility of detecting inferences involving raw sensor data, we have implemented RICE-Sy using the probabilistic logic programming language ProbLog 2 [28], to model and reason on the inference channel of our case study. This language has logical mechanisms similar to the Prolog language to reason about truth values and allows the definition of probabilistic distributions over atoms. The choice of ProbLog is motivated by the possibility of implementing our inference channel representation (e.g., the random variable $X_{1}$ using *annotated disjunction*). Moreover, we plan in the future to incorporate probabilistic dependencies related to personal data, similarly to what Chen et al. [12] have proposed, in order to have an InfDS handling inference channels using the *Distributed Dependency Strategy* (DDS) [29] on personal and sensor databases. The OpQHL is implemented as a MySQL database and the two filtering modules are SQL queries. Finally, the consolidation is implemented in Python as well as the general logic which links all the modules together.

At this stage, the system processes every issued query. Hence, when an inference attack occurs, it means that a user has gathered all the required knowledge to satisfy the constraints of a known inference channel. Hence, at this stage, evaluating the accuracy of the system is not relevant since no attack can be missed with respect to the modeled channels. In the two following sections, we focus on evaluating the impact of the proposed conceptual optimizations on RICE-Sy.

### 7.1. Evaluating the OpQHL

As described in Section 6.1, a QHL containing two MKUs which are referencing the same attribute for an overlapping time interval, e.g., $mku_{1}=\langle a,\left(1,5\right)\rangle$ and $mku_{2}=\langle a,\left(3,7\right)\rangle$, can be optimized, with respect to the size, into the OpQHL containing only mku=〈a,(1,7)〉. This way, the OpQHL provides a size gain compared to the initial QHL, while keeping track of the same user knowledge.

In order to evaluate the impact of the OpQHL, the only relevant metric is to consider the percentage of size gain obtained via the consolidation of MKUs. To do so, we first generate the MKUs. Each generated sequence contains one MKU for each of the 21 attributes in our case study, see Section 3. The MKUs in a sequence do not temporally overlap MKUs in another sequence. A total of 10 sequences are generated, resulting in a default dataset of 210 MKUs. Here, it is important that the same set of MKUs are considered in both the QHL and the OpQHL in order to measure a relevant size gain. Hence, the number of MKUs is chosen arbitrarily. To consider the size gain of the OpQHL, with respect to the number of consolidations, we generate a total of 11 datasets based on the default one. For each of those datasets, we increase the value of the threshold parameter $P_{c}\in \left[0,1\right]$ by a step of 0.1, then for each sequence in the default dataset and each MKU, we randomly decide if the MKU will be consolidated. When it is the case (i.e., $rand\left(\right)\geq P_{c}$), we replace the selected MKU with two overlapping MKUs referencing the same attribute. For each resulting dataset, we insert it into the QHL. Then, using the Algorithm 2, we compute the corresponding OpQHL, and we finally measure the gain of size provided by the OpQHL. We expect that the more MKUs can be consolidated, the better the size gain will be. More specifically, when no MKUs can be consolidated then |OpQHL|=|QHL|. When most MKUs can be consolidated, then |OpQHL|<<|QHL|.

We observe in Figure 6 that the OpQHL size ranges from being equal to the QHL size when no consolidation occurs, to a size reduction of 50% when each attribute is consolidated in each batch. Since it is unlikely that each of the queried MKUs are overlapping, a more realistic situation would be to consider the case where $P_{c}=0.5$. In this situation, the OpQHL leads to a size gain of 33.6% compared to the initial QHL. Hence, the more users are issuing queries, the higher the size gain is.

### 7.2. Evaluating the Filtering Steps

The second conceptual optimization aims to filter the OpQHL in order to provide the consolidation and the detection modules (see Section 6.2 and Section 6.3, respectively) with the minimal set of MKUs to consider for the currently processed query.

In order to evaluate the impact of those two steps, we measure the evolution of the computation time for both the detection and the consolidation modules, with and without the filtering modules. We focus on this specific metric due to the requirement of detecting inference attacks at query-time. To measure the evolution of the computation time, we generate an increasing volume of MKUs inserted into the OpQHL. The dataset is composed of 19 queries, containing each 500 MKUs, for a total number of MKUs equal to 9500. We assume that no inference attack occurs in the dataset in order to control the quantity of MKUs stored in the OpQHL, since when a query leads to an inference it is not tracked, see Section 5. Thus, for each query, at most 500 new MKUs are inserted into the OpQHL after processing the query. Each query in this dataset is provided as an input to RICE-Sy and the computation time of the consolidation module and the detection module are measured in order to assess the query-based filtering module and the search space filtering module, respectively. Those two temporal measures are performed once with the two filtering modules disabled and once with the two filtering modules enabled.

In the case where the two modules are disabled, we provide to the system the first 10 queries of the dataset only, since the computation of the detection quickly grows. We observe that the computation time of each module is quadratic, cf. Figure 7a,c, which corresponds to the computed theoretical complexity. We explain the difference in the computation time between the consolidation and detection by the usage of backtracking in the latter. The detection goes through the search space of $P(Q^{u}\cup QHL\left(u\right))$ (i.e., the powerset of the set of MKUs received as input, see ❸ in Figure 4) for each modeled inference channel in order to identify if a subset of MKUs satisfies its constraints.

In the case where the two filtering modules are enabled, we perform the same measurements by inserting all the queries from the dataset. We observe a similar order of improvement in the computation time of both modules, cf. Figure 7b,d, which is once again consistent with our theoretical algorithms. For the consolidation module, the improvement is explained by the fact that only a subset of MKUs from the OpQHL are now considered with the MKUs of the query instead of processing the whole OpQHL for each query. Similarly, the detection module now has to go through a smaller search space which does not include the whole OpQHL anymore.

We finally illustrate the computation time of each module in Figure 7e and we observe that the two filtering modules (i.e., the *search space filtering* and the *query-based filtering*) are linear and have become the most expensive computation performed by our system. In conclusion to these experiments, we observed that the proposed conceptual optimization steps reduced the overhead of RICE-Sy and make it linear. Since the search space filtering takes up to 10 s, further optimization steps still remain mandatory.

However, optimizing RICE-Sy with conceptual optimization seems unlikely to yield more improvements due to the highly frequent queries on data streams. Hence, in order for our InfDS to be used at query-time, it must perform the detection under an acceptable time, especially to fit the context of highly frequent queries on data streams. Hence, in the following section we demonstrate the possible improvement of the detection time via two mechanisms that reduce the volume of data process by RICE-Sy.

The continuous generation of sensor data leads users to issue queries at a high frequency to obtain up to date measurements. This result in a constant increase in the QHL size, which leads the two filtering modules of RICE-Sy to become more and more computationally expensive, as observed in Figure 7e. The first approach aims at partitioning the QHL into smaller search space based on logical criteria so that when RICE-Sy process a query, the two filtering modules only process a part of the QHL thus reducing the computation time.

The second approach focus on reducing the detection performed at query-time. In fact, based on the premise that most of the users of such database systems are not potential attackers, the inference detection at query-time should concern only users identified as suspicious. In the following, we present an architectural extension of our system which implements this idea based on this assumption.

## 8. Strategies to Partition the QHL

The high frequency of queries issued to sensor databases leads to huge quantities of MKUs stored in the QHL. To perform the detection, RICE-Sy retrieves MKUs obtained by a user for a specific time window (i.e., $\left(t_{b},t_{e}\right)$). To efficiently perform the detection, the extraction of those MKUs can leverage the usage of indices on the three criteria: the user identity, the temporal beginning, and the temporal end of the window. However, a single criterion is considered while the others are searched sequentially. Indices using multiple criteria are efficient only when the query uses all of them. Therefore, partitioning the QHL and finding the right parts allows a direct access to the selected MKUs and a filtering of a smaller search space compared to considering the whole QHL. The objective of the strategies is to partition the QHL into parts containing MKUs chosen based on logical criteria. Hence, when a query is processed by RICE-Sy, the criteria are used to retrieve the part of the QHL which must be considered for the detection. As illustrated in Figure 8, the *selector* module enables the filtering modules in the pre-processing stage to correctly target the relevant partition. The challenge of this approach is to base strategies on criteria that most efficiently reduce the search space of the filtering modules, while maintaining an easy procedure to select the suitable part of a QHL for a given query. For instance, a good criterion avoids situations where the selector module has to insert MKUs into two partitions, which can result in the duplication of data. In the following, we present, formalize, and evaluate a partitioning strategy to demonstrate the potential of RICE-Sy when endowed with this mechanism.

### 8.1. User-Oriented Strategy

The QHL is considered as a single search space storing MKUs of users with different behavior:(i)users which are querying huge quantities of MKUs which have more impact on the computation time of the filtering modules than(ii)users issuing queries from time to time, thus obtaining less MKUs.

The objective of this strategy is to partition the QHL into a main part containing the MKUs of the second kind of users and parts affected only to users of the first kind, as depicted in Figure 8. Having a main partition allows the QHL to create additional partitions only for users which need them. This results in a more balanced search space considered by the filtering modules. The partition criterion of this strategy is the *quantity* of MKUs obtained by a user.

When a user u∈U has obtained a quantity of MKUs reaching a threshold $t_{q}$, i.e., $\left|QHL\left(u\right)\right|\geq t_{q}$, she obtains her own partition in the QHL, denoted $QHL_{u}$. All the MKUs associated to *u* in QHL are moved into $QHL_{u}$, i.e., $QHL_{u}=QHL\left(u\right)$ and $QHL=QHL-QHL_{u}$. In this strategy, the QHL is defined as a set of partition $QHL=QHL_{main}\cup \left\{QHL_{u_{1}},\ldots ,QHL_{u_{n}}\right\}$, where $QHL_{main}$ represents the partition containing MKUs of users which have not been partitioned, and $QHL_{u_{i}}$ representing the partition created for the user $u_{i}$. The partitions associated to a user can be retrieved by the filtering modules thanks to the following function: selector:U↦MK, where $selector\left(u\right)=QHL_{u}$ if $\exists \;QHL_{u}\in QHL$ otherwise $QHL_{main}$. Later on, when the same user *u* issues a query, ❶ Figure 8, the filtering modules of RICE-Sy will consider only the search space of $QHL_{u}$ thanks to the *Selector* module.

### 8.2. Experiment: Evaluating the User-Oriented Strategy

To demonstrate the benefits of partitioning the QHL, we evaluate in this section the user-oriented strategy by measuring the evolution of the detection time of RICE-Sy. Like for the experimentation of the conceptual optimizations in Section 7.2, we focus on this metric to evaluate the proposed optimization with respect to our main requirement which is to detect inference attacks at query-time. We perform this evaluation by measuring this evolution with and without the strategy on the same dataset. For this experimentation, we generate a dataset *D* containing *m* queries issued by *n* users. The strategy is triggered when the median detection time reaches at least $t_{t}$ when processing a query and the quantity of MKUs obtained by the user is above the threshold $t_{q}$. At the end of the evaluation, we observe a decrease in the time at the level of a user when she obtains her own part of the QHL. In this case, the filtering modules have a smaller search space to consider. Globally, we note that the strategy allows the QHL to balance the search space between QHL parts affected to users having large quantities of MKUs and the main part containing users with a few MKUs. The resulting median detection time noticeably decreases with the strategy the more queries are processed by RICE-Sy. All measures have been performed on a VMware instance, using an Intel(R) Xeon(R) Gold 6126 CPU @ 2.60 GHz, and 16 GB of RAM.

Figure 9 shows the result of the evaluation performed for the user-oriented strategy, where *D* contains 19,003 queries divided over 10 users (for a total of 188,793 MKUs), the threshold is set to 125 ms and 700 MKUs, for $t_{t}$ and $t_{q}$, respectively. In Figure 9a,b, each point corresponds to the detection time of RICE-Sy when processing a query issued by user $u_{2}$, respectively, $u_{8}$. The orange dashed vertical line shows when the MKUs of the concerned user are moved from the main part to her own part of the QHL. The dark blue filled points on the right side of the orange line corresponds to the detection time when extracting MKUs from the smaller search space of both users’ new QHL part. The hollow red points correspond to the detection time without the user-oriented strategy. We observe that the detection time noticeably decreases after the partitioning due to the smaller amount of MKUs that RICE-Sy has to consider.

Figure 10 compares for the same dataset *D*, the median detection time with (bottom curve) and without (top curve) the partitioning strategy, considering the total number of queries issued by all the users. We choose to display the median in order to show the global evolution of the detection time. With partitioning, RICE-Sy is able to process the whole dataset in a shorter amount of time than without, hence the shorter line. We also observe, that each partitioning of the QHL for a user (depicted by the orange rectangles) further flatten the curve compared to the detection time without the strategy. The median time of the last query in the dataset is equal to 875 ms and 319 ms, without and with the strategy, respectively, resulting in an improvement of the detection time of 63% for this dataset.

When considering the QHL as a single partition, the filtering modules go through all MKUs obtained by all users. Incidentally, the more queries have been processed by RICE-Sy, the larger the QHL. On the other hand, the quantity of MKUs is smaller after having created a partition, the filtering modules have a smaller search space to consider for future queries, hence the detection time of queries issued afterwards decreases. Each partition also reduces the size of the main part of the QHL which also reduce filtering time. Consequently, at the global scale, the decrease in the computation time corresponds to the sum of the time gains of each created partition. However, when users continue to query sensor data after having obtained their own QHL part, the search space will continue growing. Hence, the user-oriented strategy is not sufficient to improve the median detection time on the long term, RICE-Sy must be equipped with complementary strategies to further reduce the overhead of the detection at query-time. Moreover, faced with the high frequency of queries to process, external optimizations should be provided to the detection system to lighten the computation performed at query-time. In the following, we describe an architectural extension for RICE-Sy.

## 9. H-RICE-Sy: A Hybrid Alternative to RICE-Sy

The second promising approach to reduce RICE-Sy’s impact on the query response time is based on the assumption that most of the users issuing queries do not attempt to perform inference attacks. Only few users identified as *malicious* aim to exploit inference channels involving raw sensor data (i.e., the dishonest users), while most of them are genuine. In other words, most of the queries processed at query time does not result in an inference detection, while taking time to be enforced due to the huge amount of stored knowledge. However, all the detection should be done to maintain the user knowledge up-to-date to prevent a further behavior change. To reduce the overhead of performing this detection, we propose H-RICE-Sy where queries issued by genuine users are processed offline (i.e., asynchronously with respect to the query-time), for example at night after the queries arrived. To do so, we make the assumption (The profiling module is an ongoing work. Its description is out of scope of this paper.) that a profiling system based on the user history assigns profiles to the users.

The challenges associated with this approach are the identification of the features used to profile users and the frequency at which those profiles must be updated. Multiple features can be observed in the querying behavior of users, e.g., if they follow querying patterns related to selected attributes or the time at which queries are issued. The profile associated to users must be as precise as possible in order for H-RICE-Sy to perform the best decision. This implies that the profile must be updated at a high enough frequency to reflect potential behavioral changes. Yet, this frequency must be set while considering the fact that all queries are not processed at the same time (i.e., online vs offline). Hence, either the update is performed at a high frequency with the risk of missing changes of users that are processed offline or with a slower frequency while considering all the queries issued during the last day, for instance.

The architecture of H-RICE-Sy is based on top of two instances of RICE-Sy that share the user knowledge and process the detection synchronously by the online module (vs. asynchronously by the offline module). The workflow of H-RICE-Sy is depicted in Figure 11 as follows: when receiving ❶ a query $Q^{u}$ issued by a user *u*, the first module of our system, the *dispatcher* module, determines according to the user profile (i.e., malicious/genuine) if the detection will be made online/offline. In case *u* is considered as dishonest, $Q^{u}$ is processed online ❷ and the detection result is notified to the protected DSMS ❸, in the same setting as for RICE-Sy. If the profile of user *u* is genuine, then the dispatcher module will store $Q^{u}$ in a temporary buffer and notifies the protected DSMS ❸ that the query does not lead to an inference attack. At predefined fixed periods, the offline detection is performed on the buffered queries of each user ① in order to keep the QHL up to date.

### 9.1. The Trade-Off Issue of H-RICE-Sy

Although considering the user’s profile allows the system to reduce the number of queries that have to be analyzed at query-time, it opens up the possibility of detecting inference attacks after having answered the query. Once detected, those attacks can not be prevented anymore since the queries have already been answered. The challenge is thus to determine how to minimize both:The impact of the detection at query-time (i.e., the time taken by the dispatcher to perform the decision ❷ plus the time ❸ to either perform the detection online or to buffer the query) measured by the median online detection time when processing queries.The number of inferences detected when processing the buffered queries offline ①.

The trade-off depends on the three following parameters: the acceptable median online detection time, the total volume of queries to process online, and the ratio of queries issued by malicious users. Thus, under some condition, H-RICE-Sy is capable of processing online all queries issued by malicious users without having to start minimizing both metrics. Evaluating the limits of H-RICE-Sy thus start by assessing this condition when several parameters are varying. In the following section, we first evaluate our proposal to identify the maximum volume of MKUs that H-RICE-Sy can process under several conditions, while avoiding having to process users that have been identified as malicious offline. Then, we compare the evolution of the median detection of RICE-Sy and H-RICE-Sy to assess the improvement provided by our proposed optimization.

### 9.2. Experiments: Maximum Acceptable Amount of Online Detection and Comparison of the Median Detection Time

In this section, we evaluate what the maximum volume of queries, with respect to an acceptable median detection time is, that can be processed online (i.e., queries related to malicious users) considering different ratios of malicious and genuine users. This experiment is realized with the same setting and using the same metric as described in Section 8.2. The acceptable detection time is instantiated to 80 ms accordingly to the explanations provided at the beginning of Section 6.

Intuitively, more genuine users should lead to fewer queries being processed online. The objective of the first experiment on the hybrid architecture is to observe how the median online detection time changes when processing a fixed amount of queries, distributed at different ratios between malicious and genuine users. To do so, we create 11 datasets (i.e., to cover the range from no malicious users to only malicious users) representing synthetic queries, containing a similar quantity of MKUs referencing the 21 attributes of our case study MHEALTH. Each dataset has a different ratio of queries issued by users labeled as genuine or malicious. For instance, the dataset with a ratio of 10% contains 10% of queries issued by malicious users and 90% of queries issued by genuine users. Hence, a dataset is built for each step of the ratio (from 0% to 90%, with a step size of 10%) and contains 50 users having a similar volume of queries. More specific, each user has 22,000 MKUs divided into 6000 queries. In Figure 12, we can observe the median online detection time that is measured for each configuration. As expected, we see that the higher the ratio of online processed queries (by users considered as malicious), the higher the median online time. Moreover, we observe that, as long as the proportion of malicious users is below 78%, for the considered volume of MKUs, the median online detection time stays under the acceptable detection time. It is especially true if we consider a low number of malicious users, since for the ratio 10%, 20%, and 30% the median online time does not increase significantly.

In the second experiment, we aim to identify the maximum volume of MKUs which can be processed online while staying below the acceptable detection time defined in Section 6. We will consider the following ratio of malicious users: 5%, 30%, and 78%. The first two values have been chosen by following our initial assumption stating that only a few users try to perform inference attacks. The last value has been chosen as a reference, since it corresponds to the maximum ratio of malicious users that H-RICE-Sy was able to process online, while staying under the acceptable time. In order to perform this evaluation, we have once again generated datasets containing synthetic queries. The five datasets, labeled D_1_ to D_5_, contain different volumes of MKUs, see the table in Figure 13.

The median growth rate of the volume between the datasets is approximately equal to 0.45. The growth rate of the median online detection time for the ratio of 5%, 30%, and 78% is approximately equal to 0.13, 0.18, and 0.27, respectively. We observe that the higher the ratio of volume processed online is, the larger the growth rate of the median becomes. Focusing on the two realistic ratios (i.e., 5% and 30%), we can estimate the amount of volume that can be processed online by using the observed growth rate. For 30%, we deduce that for a volume of approximately 1.9 × 10^6^ MKUs, the median online detection time is 85 ms, just above the threshold of the acceptable time (i.e., 80 ms). In order to evaluate this estimation, we have generated a sixth dataset, D_6_, containing a volume of MKUs of the same order of magnitude as the estimated volume. We have not performed measurements for 78% since we are focusing on the more realistic ratios. We can observe that for 30%, a volume of 1.2 × 10^6^ MKUs leads to a median online detection time of 85 ms. We conclude that our estimation is adequately accurate to provide relevant insights. For 5%, we deduce that for a volume of approximately 8.6 × 10^6^ MKUs, the median online detection time reaches 85 ms. We need to continue the evaluation on larger datasets in order to verify that we reach the estimated volume for the ratio of 5%.

Finally, we compare the improvement provided by H-RICE-Sy with respect to the detection time, using RICE-Sy as the baseline. To do so, we measure for both systems the evolution of the median detection time when processing a small and a medium sized dataset: D_1_ and D_3_, see Figure 13, in order to see if the improvement provided by H-RICE-Sy scales with the amount of processed queries. For both dataset, 30% of the MKUs are affected to malicious users, thus they are processed online by H-RICE-Sy. Choosing 5% or 30% both match our assumption that only a small quantity of users are malicious, however we choose the second value to consider a *worst* situation where more queries have to be processed online. The peaks at the beginning of each curve depicted in Figure 14 are present since the computation of the median detection time is performed on only the first measurement of the first query. Figure 14a shows that, as expected, the possibility provided by H-RICE-Sy to process only queries of malicious users online reduces the median detection time. Moreover, we also observe that the constant increase in the QHL leads to a constant increase in the median detection time which is similar for both RICE-Sy and H-RICE-Sy. We can observe in Figure 14b that the improvement of H-RICE-Sy scales with the size of the dataset.

In the first experimentation, we have observed the limits under which the trade-off can be avoided. With a small number of malicious users (5%) and a short detection time ( 80 ms), H-RICE-Sy is capable of processing a high amount of MKUs without having to deploy further strategies to satisfy the trade-off. Although the system continues to process queries, or if it changes its parameters, it will always reach the maximum volume which can be processed. Therefore, as a future perspective, the H-RICE-Sy must be endowed with a mechanism deployed in the *dispatcher* module which detects when the limit is reached by the system. This mechanism then applies strategies, such as probabilistic heuristics, in order to process some queries of malicious users offline until the median online detection time is minimized under the acceptable time, i.e., metric 1 in Section 9.1. The challenge of those strategies is two fold: (i) it must identify the malicious users that have the smallest probability to perform inference attacks while their queries are dispatched offline, i.e., metric 2; and (ii) it must be as fast as possible since it impact the online detection time.

## 10. Discussion

All proposed optimizations are aimed at reducing the overhead of RICE-Sy thus the metric we considered is the detection time at query-time. In paradigms such as Edge or Fog Computing, devices are deployed near sensors to provide limited power and storage capacity [30]. An instance of our system deployed on those devices has to cope with those limitation, hence other relevant metrics would be to observe the evolution of the memory usage, as well as the energy consumption of RICE-Sy and the proposed optimizations.

The two mechanisms (the partitioning strategy in Section 8 and H-RICE-SY in Section 9) demonstrate the potential of RICE-Sy to reach acceptable detection times in the context of continuous sensor data generation. However, the efficiency of H-RICE-Sy relies mainly on the precision of the profiling system. Classifying users based on their querying behavior is pertinent since the system has access to those information. Yet, considering only those information is not sufficient to discriminate between a user with a genuine intent and a malicious user. For instance, from the perspective of RICE-Sy, when a user has queried enough knowledge to exploit an inference channel, an attack is detected even so the user is not aware of an existing dependency. Although this limitation is inherent to InfDSs which model both knowledge of users and inference channels, profiling users based on more features than the one from the querying behavior can limit the wrong decision performed in such situations.

RICE-Sy and the two proposed mechanisms fill the gap of inference detection against attacks on raw sensor data to obtain personal data. The obtained personal data by itself can be non-sensitive in most situations and does not breach someone’s privacy, but combined with external data from other databases, it can help to reach more sensitive data, escaping the vigilance of InfDSs which act locally on the databases. A complete use case illustrating this issue is shown in [11]. In this use case, distinct services are collecting both personal and sensor data of their customers. Authorized access to those data is provided by services to external users, with respect to the customers’ consent. Dishonest users, however, can obtain indirectly access to sensitive personal data stored by the service by performing inference attacks that leverage the *Distributed Dependency Strategy* (DDS) [29].

## 11. Conclusions

In this paper we have presented RICE-Sy, a novel inference detection system suitable to capturing inference attacks on raw sensor data. RICE-Sy is based on RICE-M (Raw sensor data-based Inference ChannEls Model), an extensible model that represents both inference channels created by new learning algorithms on sensor data and the corresponding knowledge capturing metadata on the queried sensor data. A proof of concept of RICE-Sy, considering the MHEALTH inference channel, has been realized and validates the possibility of detecting inference attacks involving raw sensor data.

Sensor data become large very quickly. As a consequence, efficiently detecting inference attacks at query-time is mandatory. To reduce the detection cost, we have first proposed a set of conceptual optimizations which aim at reducing the user knowledge to process. The obtained detection time becomes linear, although it remains greater than the acceptable time of 80 ms. In the second part of the paper, we have shown the potential efficiency enhancement of RICE-Sy, by proposing two approaches. The first consists in partitioning the knowledge gained by users into parts applying logical criteria, e.g., per user identity. The partitioning allows to target smaller search spaces in the detection algorithm. We experiment with the user identity as a discriminating criterion and observed a gain of 63%, which is a promising first result. The second approach we have investigated is a hybrid architecture of RICE-Sy named H-RICE-Sy. This new architecture is conditioned by a profiling system able to assign malicious/genuine profiles to users based on their query history. In H-RICE-Sy, only queries of malicious users are processed at query-time. H-RICE-Sy reduces the median time of detection by limiting the detection to queries issued by malicious users. Thanks to our experiments, we evaluate to which extent such a system limits the detection overhead while avoiding a posteriori inference attacks detection. The experiments show the potential room of improvement that the H-RICE-Sy can yield, since the median online detection time remains under the acceptable time of 80 ms, even for a high volume of user knowledge (≈1.2 × 10^6^ MKUs) for a ratio of 30% of malicious users, which is higher than what could be expected in real life. Based on the observed growth rates, we have estimated that for 5%, a maximum volume of approximately 8.6 × 10^6^ MKUs, obtained by malicious users, can be processed online in an acceptable time.

### Future Works

As a near future work, we plan to investigate partitioning strategies based on different logical criteria, such as the time at which selected sensor data are generated to evaluate which strategies can be combined to further reduce the detection time while keeping a logically simple selection of the partition. Concerning H-RICE-Sy, we are working on the profiling module which allows assigning profiles to users (i.e., malicious/genuine) based on their query history, in order to choose which detection (i.e., online/offline) is suitable. The second future work concerns the evaluation of H-RICE-Sy, especially in scenarios where the ratio of malicious users is important, or a small amount of malicious users are performing most of the queries. In this case, a hybrid detection is acceptable if both the number of offline detected inferences and the mean detection time remain low. We plan to refine the definition of the acceptable detection time presented in Section 6 to further take into consideration the specificities of both the detection of inference attacks at query-time and the heterogeneous nature of the collected sensor data. We plan to investigate strategies to be applied by the dispatcher module, when H-RICE-Sy cannot minimize the median online detection time below an acceptable threshold, for example by implementing heuristics, see the last paragraph in Section 9.2, which will aim to redirect queries based on profiles and the metrics. We are also working on a RICE-M extension to support spatial constraints considered in some inference channels like in [4].

## Figures and Tables

**Figure 1 sensors-22-08140-f001:**
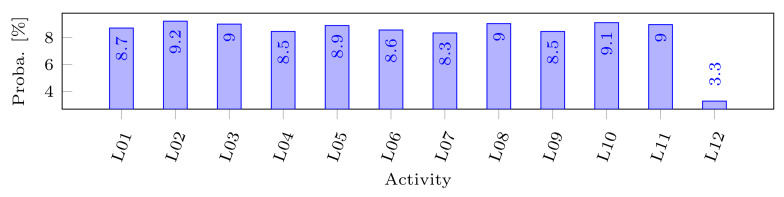
Probability distribution of classifying human activities in the MHEALTH.

**Figure 2 sensors-22-08140-f002:**
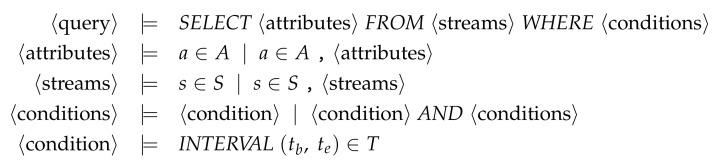
Theoretical grammar of queries issued on data streams. A summary of the symbols definition is listed in Table 1.

**Figure 3 sensors-22-08140-f003:**
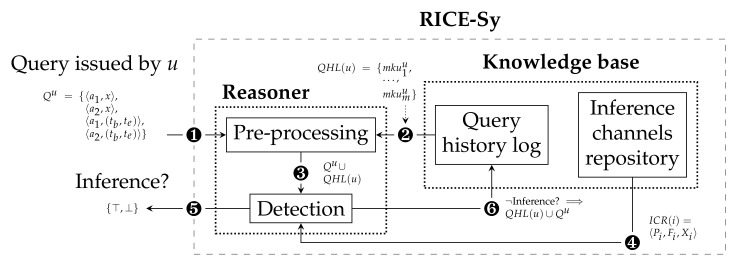
Generic workflow of RICE-Sy.

**Figure 4 sensors-22-08140-f004:**
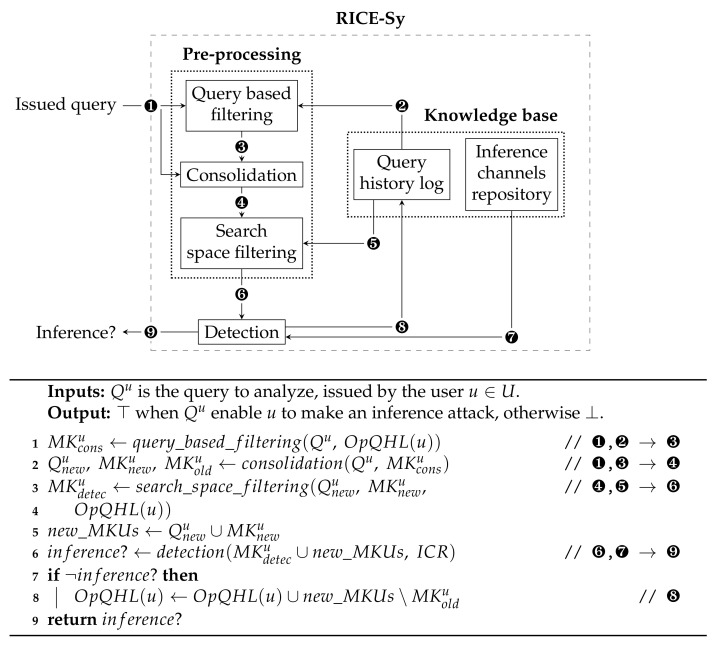
Complete workflow of our RICE-Sy with the main function.

**Figure 5 sensors-22-08140-f005:**
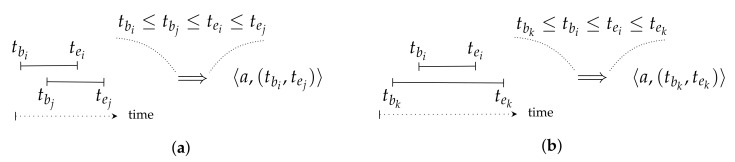
Consolidation rules for temporal intervals. (**a**) Overlapping rule. (**b**) Nested rule.

**Figure 6 sensors-22-08140-f006:**
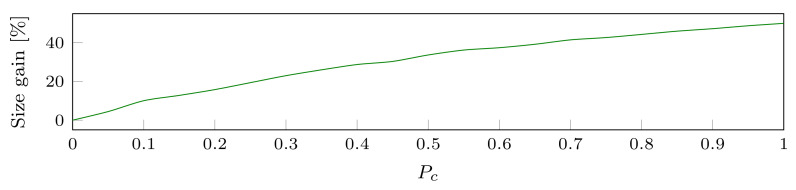
Size gain of the OpQHL compared to the initial QHL, for a varying probability $P_{c}$ of having MKUs that can be consolidated. Where $P_{c}=0$ means that no consolidation is happening and $P_{c}=1$ means that each MKU in the dataset can be consolidated once.

**Figure 7 sensors-22-08140-f007:**
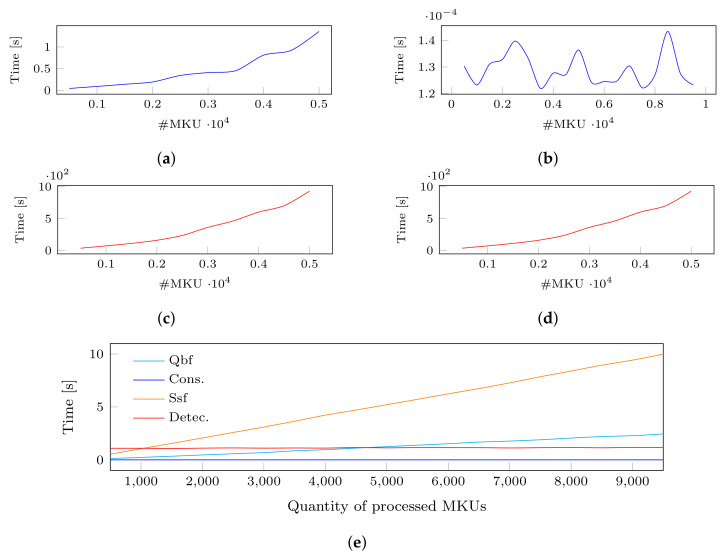
Evolution of the computation time of the consolidation module (Cons.) and the detection module (Detec.), without (¬) and with (∧) the query based filtering (*Qbf*) and the search space filtering (*Ssf*) enabled, for every processed query of 500 MKUs. (**a**) Cons. ¬ filtering. (**b**) Cons. ∧ filtering. (**c**) Detec. ¬ filtering. (**d**) Detec. ∧ filtering. (**e**) Evolution of the computation time of each module in RICE-Sy.

**Figure 8 sensors-22-08140-f008:**
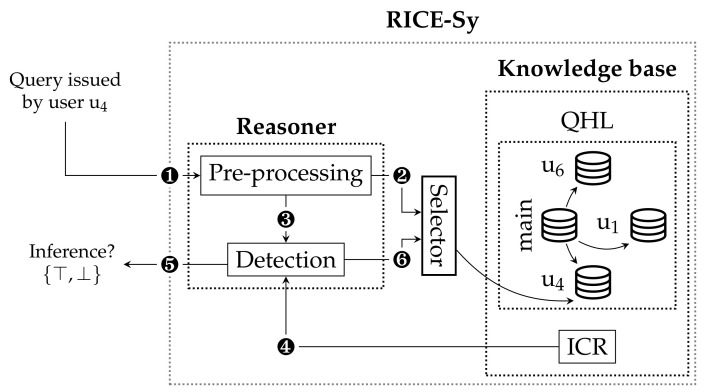
User-oriented partitioning of the QHL when a query issued by u_4_ is processed by RICE-Sy.

**Figure 9 sensors-22-08140-f009:**
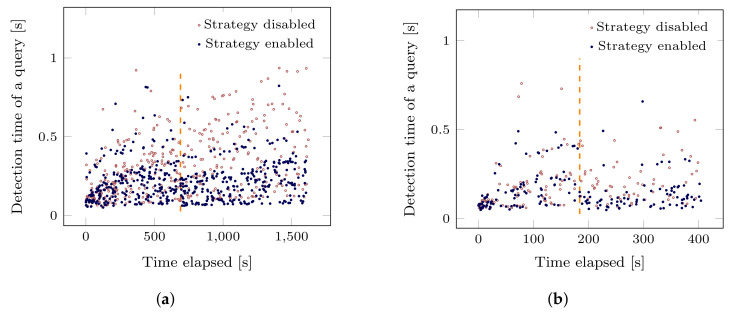
Detection time of $u_{2}$’s and $u_{8}$’s queries with and without the user-oriented partitioning strategy. The orange dashed line depict the time at which the user has been affected its own part of the QHL. (**a**) $u_{2}$’s queries. (**b**) $u_{8}$’s queries.

**Figure 10 sensors-22-08140-f010:**
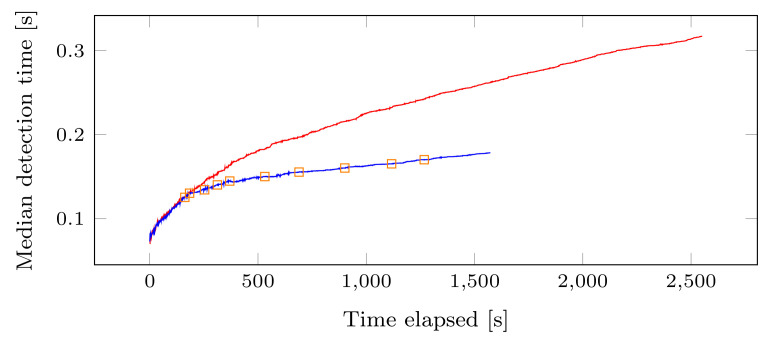
Evolution of median detection time of RICE-Sy with (bottom) and without (top) the user-oriented strategy. The orange squares show when the QHL is partitioned for a user. Only the first 5000 measurements are displayed for readability reasons.

**Figure 11 sensors-22-08140-f011:**
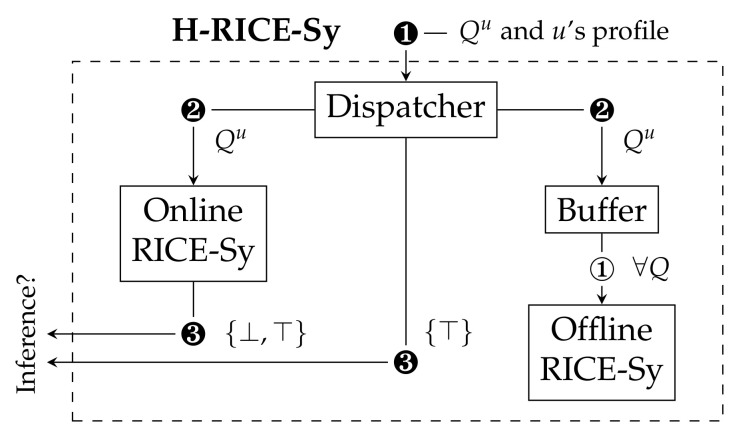
Workflow of the hybrid RICE-Sy (H-RICE-Sy). The arrows labels with a black or white backgrounds mean that the action is performed online or offline, respectively.

**Figure 12 sensors-22-08140-f012:**
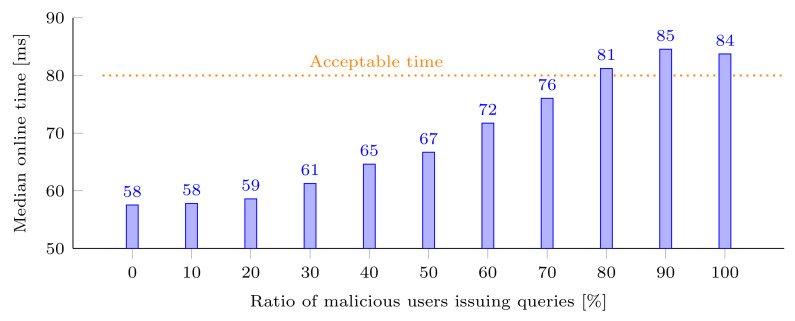
Median online time for different ratio of queries issued by malicious users among a total of 50 users. Compared with the acceptable detection time defined in Section 6.

**Figure 13 sensors-22-08140-f013:**
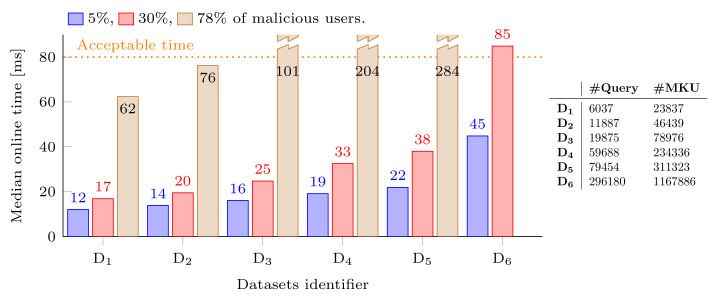
Evolution of the median online time on datasets with fixed volume of MKUs and varying ratio of queries affected to malicious users, and thus processed online. The table on the right side describes the volume of queries and MKUs for each dataset.

**Figure 14 sensors-22-08140-f014:**
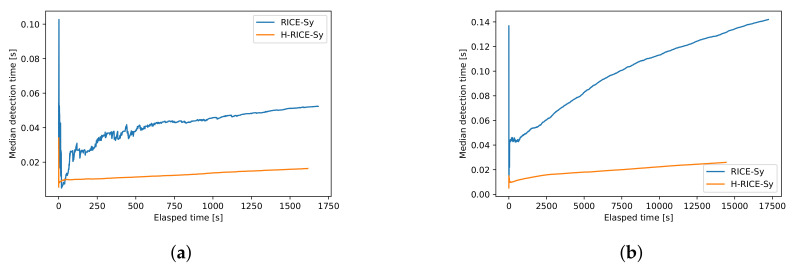
Comparison of the median detection time evolution between RICE-Sy and H-RICE-Sy on D_1_ and D_3_. (**a**) Measurement where 30% of MKUs in D_1_ are affected to malicious users. (**b**) Measurement where 30% of MKUs in D_3_ are affected to malicious users.

**Table 1 sensors-22-08140-t001:** Symbols defined to model user knowledge in RICE-M.

Symbol	Description
S	Queryable data streams
$A={⋃}_{s\in S}A_{s}$	Attributes within data streams
$(t_{b},t_{e})\in T$	A time duration
〈a∈A,c∈T〉∈MK	Metadata Unit Knowledge (MKU)
Q⊆MK	MKUs related to a query
U	Users querying data streams
QHL:U↦MK	Query History Log of users

**Table 2 sensors-22-08140-t002:** Symbols defined to model inference channels in RICE-M.

Symbol	Description
I	Identifiers of inference channels
〈P∈P,F∈F,X∈X〉∈ICD	Inference channels
ICR:I↦ICD	Description of inference channels
P	Patterns defining MKUs shapes for inference
	channels in ICD
F	Constraints of inference channels in ICD
A${}_{inf}$	Inferable attributes of inference channels in ICD
X	Probability distribution of the inferable attribute
	of inference channels

## Data Availability

The MHEALTH dataset can be found here: https://archive.ics.uci.edu/ml/datasets/MHEALTH+Dataset (accessed on 9 September 2022).

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
