# Peer review of "Detecting Inference Attacks Involving Raw Sensor Data: A Case Study"

_sensors, 2022, doi:10.3390/s22218140_

Round 1

Reviewer 1 Report

The paper proposes a workable method for detecting sensor inference attacks. It is novel to apply Inference Detection Systems (InfDSs) to the sensor domain to detect potentially occurring inference attacks. The approach can address the possibility of threats causing an invasion of users' private data. The paper also proposes improved methods for optimizing time complexity.

However, some flaws must be addressed before it can be considered for publication.

(1) Because the paper only reflects how it is used, it is suggested that the Inference Channels repository's construction details be added.

(2) The content of several figures, including Figures 6, 7, and 9, is missing.

(3) In the evaluation section, it should be clear what metrics were chosen and why, and additional metrics should be suggested.

(4) More experiments, including some comparison experiments, are required.

(5)The papers cited are a little out of date. The vast majority of references are from prior to 2018, with only a few from 2019 and 2020. There is no analysis of relevant work between 2021 and 2022.

Reviewer 2 Report

The review comments on this manuscript are listed as follows:

1. Some abbreviations should have their full texts when they appear first in the manuscript.

2. In line 134, the “NP-Complet problem” should be “NP-Complete problem”.

3. The authors should specify the literature source; moreover, there is not a paragraph mentioned Figure 1 in the manuscript.

4. There exist some mistyping vocabularies in the manuscript.

5. The authors should explain the reason why they used the MHEALTH dataset as a case study in their study.

6. Figure 2 should have a legend to explain the used symbols. Moreover, the authors should have a table to list the definitions of all the symbols in Subsection 4.2, Subsection 4.3, and Section 5.

7. The authors should explain the meanings of the solid arrow lines and the dashed arrow lines shown in Figure 3, Figure 5, and Figure 8.

8.    The authors should resize Figure 6 and Figure 7 to let readers can examine the whole figures shown in Figure 6 and Figure 7.

9.    The authors should have a legend for Algorithm 1, Algorithm2, and Algorithm 3.

10. There is not a paragraph mentioning Algorithm 3 in the manuscript.

11.   Figure 9(a) does not show an orange line; the authors should revise Figure 9(a).
